# Unsupervised Exploration with Deep Model-Based Reinforcement Learning

## Abstract

Reinforcement learning (RL) often requires large numbers of trials to solve a single specific task. This is in sharp contrast to human and animal learning: humans and animals can use past experience to acquire an understanding about the world, which they can then use to perform new tasks with minimal additional learning. In this work, we study how an unsupervised exploration phase can be used to build up such prior knowledge, which can then be utilized in a second phase to perform new tasks, either directly without any additional exploration, or through minimal fine-tuning. A critical question with this approach is: what kind of knowledge should be transferred from the unsupervised phase to the goal-directed phase? We argue that model-based RL offers an appealing solution. By transferring models, which are task-agnostic, we can perform new tasks without any additional learning at all. However, this relies on having a suitable exploration method during unsupervised training, and a model-based RL method that can effectively utilize modern high-capacity parametric function classes, such as deep neural networks. We show that both challenges can be addressed by representing model-uncertainty, which can both guide exploration in the unsupervised phase and ensure that the errors in the model are not exploited by the planner in the goal-directed phase. We illustrate, on simple simulated benchmark tasks, that our method can perform various goal-directed skills on the first attempt, and can improve further with fine-tuning, exceeding the performance of alternative exploration methods.

## 1 Introduction

Many mammals, including humans, are incapable after birth of providing for themselves for extended periods of time (years in the case of humans). This behavior is in stark contrast with other species that are born with complex motor skills already available to them. One hypothesis is that this forced developmental period provides a safe environment for the infant to perform unsupervised (i.e., task-agnostic) exploration in order to acquire more flexible skills and enhanced adaptation capabilities that will prove useful in the long term. This hypothesis is at least partially supported by the observation that reduced exploration in the early periods (7-9 months) of infancy is correlated with long-term poorer cognitive and language outcomes (Ruff et al., 1984; Muentener et al., 2018).

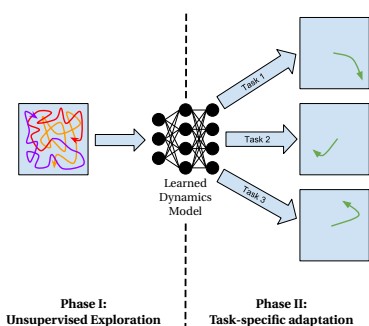

Figure 1: The unsupervised model-based RL setting. An unsupervised phase is used to build a predictive model using our proposed exploration criterion. When a task is then specified, the agent can immediately perform with no further training.

From a machine learning perspective on reinforcement learning, this hypothesis raises intriguing questions. The predominant formalization of RL is concerned with solving a single task. Is learning to solve a single task "in the void" without any context and prior knowledge truly the right benchmark to develop intelligent and efficient algorithms? Can human-like learning capabilities be attained through iterative progress in such simplified settings? Or can we formulate a problem statement that more closely

matches the one outlined above, where an unsupervised critical period is used to acquire knowledge about the world, and that knowledge is then used for goal-directed task acquisition later?

Guided by these overwhelming questions, in this paper we do a timid step towards better understanding the importance of unsupervised exploration and reinforcement learning. To do so, we consider a setting composed of a first *unsupervised exploration phase* where the agent can interact with the environment – but without having any specific task defined – and a second phase which require the agent to quickly adapt to solve multiple tasks not previously known. We then address a specific piece of the critical questions: if we are to use an unsupervised exploration phase to learn about the world, what sort of knowledge about the world should we retain that will be helpful for then achieving specific goals? The particular hypothesis that we explore in this regard is that model-based reinforcement learning offers a promising solution, if it is augmented with an appropriate unsupervised exploration strategy. In contrast to model-free reinforcement learning, which learns task-specific policies or value functions, model-based reinforcement learns to predict the future – specifically, to predict the next state from the previous state and action. This predictive model can then be used to optimize for near-optimal behavior by planning or offline policy optimization. Model-based reinforcement learning offers an appealing approach to unsupervised exploration: explore the environment to acquire a good model, in effect learning how the world works, and then simply use this model to achieve new user-specified tasks. In contrast, model-free unsupervised reinforcement learning methods require considerable additional machinery to extract useful knowledge from unsupervised exploration.

While this basic principle is simple and well known, to our knowledge no prior work has proposed an unsupervised approach to exploration in model-based reinforcement learning with expressive deep neural network models. This formulation raises several challenges. First, model-based RL with deep neural networks is inherently challenging because high-capacity models have a tendency to overfit in the model-based RL setting, leading to poor performance. The optimization over actions or policies exacerbates this issue, since a model that overfits to data will be easy for a strong optimizer to "exploit" and discover erroneously optimistic outcomes. Second, unsupervised reinforcement learning requires an effective exploration procedure to achieve good coverage of the state, and while exploration for model-free RL has been studied extensively, as we discuss in the following section, exploration for model-based RL has been studied substantially less.

In this work, we use model uncertainty to address both of these issues. By formulating an uncertainty-aware deep neural network dynamics model, we show that we can both obtain a readily usable exploration criterion for the unsupervised training phase and produce a model-based RL procedure that can remain robust to "exploitation" when presented with goals at test-time. Our primary contributions are the formulation of the unsupervised deep model-based RL learning procedure, and the introduction of a practical algorithm for tackling this problem. Our experimental results show that our method can provide excellent performance in terms of achieving new goals, even without any additional training beyond the unsupervised exploration phase, and can substantially improve performance through additional finetuning. We compare to alternative exploration heuristics from the model-free RL literature, as well prior baseline methods that do not perform explicit exploration.

## 2 RELATED WORK

Exploration generally aims to acquire information about an environment to help an agent better optimize its reward function. A large number of exploration techniques have been proposed in the reinforcement learning literature suited for continuous state-action environments, from simple inefficient methods to complex methodical exploration strategies. Simple but computationally-cheap methods used to increase training data diversity include occasionally selecting random actions ($\epsilon$-greedy) for discrete actions, additional Gaussian noise for continuous actions, and policy parameters perturbations (Plappert et al., 2017; Fortunato et al., 2017). A disadvantage of $\epsilon$-greedy exploration is even actions that are known to have low Q-values are executed with the same probability as other actions. Exploration can instead be biased towards actions of higher expected value using Boltzman exploration or maximum entropy methods (Haarnoja et al., 2018). However the robot may still repeatedly 'explore' actions whose values it is already certain. Information value theory defines the value of exploration as the additional rewards we expect to obtain given the expected information of taking an action (Howard, 1966). In reinforcement learning, the value of exploration

(information) can be formulated with Bayesian method by first placing a prior over the space of MDPs and treating all subsequent observations as likelihoods. This Bayesian reinforcement leaning transforms the MDP *learning* problem into a POMDP *planning* problem: a POMDP whose state is a hybrid of dynamics function parameters (that are partially observed, or uncertainty) and the robot state (Duff & Barto, 2002). Whilst treating RL as a POMDP defines exploratory value in MDP environments, computing values is almost always intractable. Instead approximate myopic and 'optimistic' exploration strategies can form effective explorers in model-free or model-based RL (Dearden et al., 1998; 1999; Ghavamzadeh et al., 2015). Such frameworks reason probabilistically about plausible value functions or dynamics functions respectively. Directing exploration towards such known-unknowns yields exponentially faster learning in discrete spaces (Thrun, 1992). For continuous state-action spaces, an effective method periodically samples value function or dynamics function, acting greedily w.r.t. the sample for a period of time, achieving temporarily-extended (or deep) exploration Thompson (1933); Osband et al. (2017); Touati et al. (2018).

An alternate definition of data-efficiency for exploration is minimizing the number of episodes that the agent fails to perform within a specified bound of the optimal performance. Algorithms which address this alternate criteria are PAC-MDP (probably approximately correct Markov decisions process) methods which aim to discover 'near-optimal' policies with high-probability within time polynomial to the number of states and actions. Examples include R-max Brafman & Tennenholtz (2003), $E^3$ Kearns & Singh (2002), and delayed Q-learning Strehl et al. (2006). PAC-MDP methods provide powerful probabilistic-guarantees on the data-complexity required before asymptotic convergence to an optimal policy. However, such strong guarantees are only possible by systematic over-exploration of the complete state-space Delage & Mannor (2007); Kolter & Ng (2009).

In the context of model-based RL, uncertainty has been previously used both to encourage exploration in states of high-uncertainty (Depeweg et al., 2017), and to perform risk-sensitive control by avoiding states with high uncertainty (Buechler et al., 2018).

Whilst the aforementioned Bayesian methods are typically effective in typical RL problems where the reward function or reward samples are accessible, this paper considers the problem an initial exploration training phase that is *unsupervised* w.r.t. the reward. Hence our problem setting is more akin to using state-uncertainty-based exploration methods (Bellemare et al., 2016; Houthooft et al., 2016; Tang et al., 2017), model error (Pathak et al., 2017), and novelty-based exploration (Fu et al., 2017). However, in contrast to all of these prior methods, which are concerned either with exploring to maximize a reward on a single specific task, or else exploring with no task reward at all and without any downstream tasks (Pathak et al., 2017), our method is concerned with transfer: using unsupervised exploration to acquire knowledge about the world that can help to perform downstream tasks. We therefore employ model-based reinforcement learning, in contrast to all of these prior methods that are based on model-free RL. We show that, in the model-based RL setting, transfer is straightforward, and uncertainty-aware models directly provide a readily usable exploration criterion.

To purposefully explore a state space, model-based RL can be used to intentionally plan towards uncertainty regions. Probabilistic model-based RL learns particularly fast, being robust to overfitting to small datasets during the early stages of learning (Deisenroth & Rasmussen, 2011; Chua et al., 2018). In addition, probabilistic models are well suited to exploration problems since an agent can methodically plan actions sequences of maximal trajectory uncertainty. Such behavior yields temporarily-extended (or deep) exploration as opposed to myopic exploration which only considers what act be learned within one future transition.

## 3 UNSUPERVISED EXPLORATION WITH DEEP MODEL-BASED RL

We consider the class of learning tasks, shown in Figure 1, that have two phases:

1. A longer unsupervised exploration phase, without any reward feedback.
2. A supervised phase, where the agent is provided with a goal, and must achieve that goal either immediately or after a small amount of fine-tuning.

In order to transfer knowledge from phase 1 to phase 2, we propose to use model-based reinforcement, which learns to predict the outcomes of actions in the environment. The dynamics model

learned in model-based RL is, insofar as it is globally accurate, agnostic to the particular task, making it ideal for transferring to phase 2. However, model-based RL with high-capacity function approximators, such as deep neural network models, is prone to overfitting, which can lead to poor results in practice. Furthermore, the unsupervised phase requires an objective to ensure the learner can visit a wide variety of states in the environment. To address both of these challenges, we extend the PETS algorithm (Chua et al., 2018), which estimates model uncertainty using an ensemble of dynamics models. While PETS has been previously demonstrated to achieve good performance on purely goal-directed tasks, prior work used simple random exploration, and did not study the unsupervised reinforcement learning setting. Since a probabilistic models predict distributions over future states, we extend PETS to explore by selecting action sequences that lead to a wide distribution over possible states. Through recursive prediction of state distributions, this method can consider the fact that multiple deterministic transitions may be required to reach novel states with uncertain dynamics. Such temporally extended exploration (also called deep, or non-myopic exploration) is desirable to learn complex sequential decisions making problems like walking or running. During phase 2, we propose that our algorithm act in a greedy fashion: simply maximizing the expected rewards without further consideration of exploration. If the exploration phase was successful, this will result in good performance. If performance is insufficient, several additional episodes can provide enough data to finetune the model, as we illustrate in our experiments.

## 3.1 MODEL-BASED REINFORCEMENT LEARNING

In this section, we describe the model-based reinforcement learning (MBRL) framework and relevant notation. Framing reinforcement learning as finding an optimal policy for a Markov decision process (MDP) (Bellman, 1957), we let $s_t \in \mathcal{S}$ and $a_t \in \mathcal{A}$ denote the state of the system and action taken at time $t$, respectively. The dynamics of the MDP are governed by a probabilistic transition function $f : \mathcal{S} \times \mathcal{A} \to \mathcal{S}$ such that $s_{t+1} \sim f(s_t, a_t)$, and the task on the MDP is defined by a reward function $r(s, a)$ such that the optimal action sequence is given by

$$\arg\max_{a_1, a_2, \dots} \sum_{t=0}^{T} r(s_t, a_t) \quad \text{subject to} \quad s_{t+1} \sim f(s_t, a_t), \quad t = 0, \dots, T.$$

In model-based reinforcement learning, data from interactions with the environment is used to learn a model $\widetilde{f}$ of the MDP dynamics. This model can be used to make predictions about how the system will evolve over time when acting with respect to a policy $\pi : \mathcal{S} \to \mathcal{A}$. One can then predict the distribution over returns when acting under a specific policy, and optimize the policy being used. In the following section, we describe an algorithm adhering to this framework that is central to our method.

## 3.2 PROBABILISTIC ENSEMBLES WITH TRAJECTORY SAMPLING (PETS)

PETS is a model-based deep reinforcement learning algorithm which was recently shown by Chua et al. (2018) to achieve state-of-the-art results on several continuous control benchmark tasks. In PETS, an ensemble of neural networks whose outputs parametrize distributions is taken to be the model of environment dynamics. This choice of model is accompanied by a trajectory prediction method which uses several particles independently propagated through the model to approximate the trajectory distribution induced by an action sequence.

This choice of model and propagation method was shown to have resulted in performance rivaling that of several prior model-based and model-free RL approaches. Beyond performance gains, however, this choice of model and propagation method is amenable to exploration in that it isolates two kinds of uncertainty in the dynamics - *aleatoric uncertainty*, or inherent stochasticity in the environment, and *epistemic uncertainty*, or subjective uncertainty due to lack of data. Our method takes advantage of this separation by constructing a reward function which drives the agent towards regions in the state space of high epistemic uncertainty. By encouraging the agent to seek out states where the model has insufficient data, the global dynamics model acquired should in theory be able to model a larger part of the state space, making adaptation to specific tasks easier.

Let $\widetilde{f}_\theta : \mathbb{R}^{d_s + d_a} \mapsto \mathbb{R}^{d_s}$ denote a neural network with parameters $\theta$ whose outputs parametrize a distribution, which we take to be Gaussian. We consider this neural network to be an approximation

of the true dynamics $f$, and treat it as a mapping $(\boldsymbol{s}_t, \boldsymbol{a}_t) \mapsto \boldsymbol{s}_{t+1}$. As in PETS, we obtain a set of $b$ parametrizations $\{\theta_1, \ldots, \theta_b\}$ for $\widetilde{f}$ by training several copies of the same network from subsets $\mathcal{D}_1, \ldots, \mathcal{D}_b$ sampled with replacement from a dataset $\mathcal{D}$ of transitions seen so far in the environment. These networks are trained discriminatively by optimizing the parameters to maximize the log-likelihood of their corresponding sampled datasets.

Upon training the ensemble, PETS uses the model to plan by using a particle propagation method to estimate the distribution over trajectories induced by any action sequence. Let $(\boldsymbol{a}_t, \ldots, \boldsymbol{a}_{t+T})$ be a length-$T$ action sequence, and assume that the agent is currently at state $\boldsymbol{s}_t$. Then, PETS initializes $p$ particles $\boldsymbol{s}_t^1, \ldots, \boldsymbol{s}_t^p$, all initialized to $\boldsymbol{s}_t$, and predicts trajectories by recursively sampling the next state as $\boldsymbol{s}_{t+1}^p \sim \widetilde{f}_{\theta_{b(p)}}(\boldsymbol{s}_t^p, \boldsymbol{a}_t)$, where $b(p)$ is uniformly sampled from $\{1, \ldots, b\}$. That is, the bootstrap corresponding to each particle is chosen and fixed at the beginning of trajectory prediction. To select an action, PETS uses model predictive control, or MPC, optimizing a length-$T$ action sequence at every time step using the cross-entropy method (CEM) and selecting the first action.

### 3.3 State Uncertainty-Driven Exploration

While PETS can achieve good results on conventional goal-directed model-based reinforcement learning problems, it does not by itself provide any mechanism for exploration other than adding random noise. As we will show in our experimental evaluation, this is usually insufficient to extract sufficient knowledge about the environment during the unsupervised training phase. In this section, we propose a novel exploration method that can be combined readily with PETS to achieve good coverage of the state space and lead to good performance on downstream tasks in phase 2, even without any additional training. The key insight in this method is that PETS already provides us with the facilities for estimating the uncertainty of state transitions, and we simply need to construct a reward function to induce PETS to visit the states where it itself thinks transitions are uncertain. Since PETS plans for maximization of the reward over the entire horizon, this automatically causes it to take sequences of actions, each of which might have low uncertainty, if in the end it can reach a state with high uncertainty.

We can estimate uncertainty at a state with PETS as following. At every time step $t$, we compute the average over all the particles assigned to each bootstrap, and compute the variance $v_t$ over these computed means. This computed variance encapsulates epistemic uncertainty, since disagreement across models in ensemble is correlated with this variance. However, it is quite difficult to utilize $v_t$ directly as a measure of epistemic uncertainty, since each component of $v_t$ will have different units and magnitudes depending on how the state $\boldsymbol{s}_t$ is defined. Therefore, we divide $v_t$ by some baseline variance $w_{\text{base}}$ component-wise in order to eliminate units and allow the resulting components to be of the same magnitude relative to each other. In our case, we set $w_{\text{base}}$ to be the maximum aleatoric variance, a trainable parameter of the PETS model that is present for model prediction stability. We then set the reward at time step $t$ to be

$$ r_t = \frac{1}{d_s} \sum_{k=1}^{d_s} \sqrt{\frac{v_t^i}{w_{\text{base}}^i}} \, . $$

With our choice of $w_{\text{base}}$, we can view the reward as encouraging the agent to move towards regions where its predicted noise at a single time step is larger than what the model has seen in the training data. Pretraining is then achieved by using the reward function defined above together with the PETS algorithm described in the previous section.

## 4 Experimental Results

In our experimental evaluation, we evaluate how well our two-phase unsupervised exploration algorithm can transfer knowledge from the unsupervised phase to succeed on downstream tasks. We use the HalfCheetah OpenAI gym environment, which we modify to obtain four distinct downstream tasks: running forward to maximize lateral velocity, running backward to maximize negative lateral velocity, tumbling forward to maximize angular velocity, and tumbling backward to maximize negative angular velocity. As an example, Figure 2 shows rewards encouraging high velocities and angular velocities in both directions. We first execute the exploration phase, which consists of 60

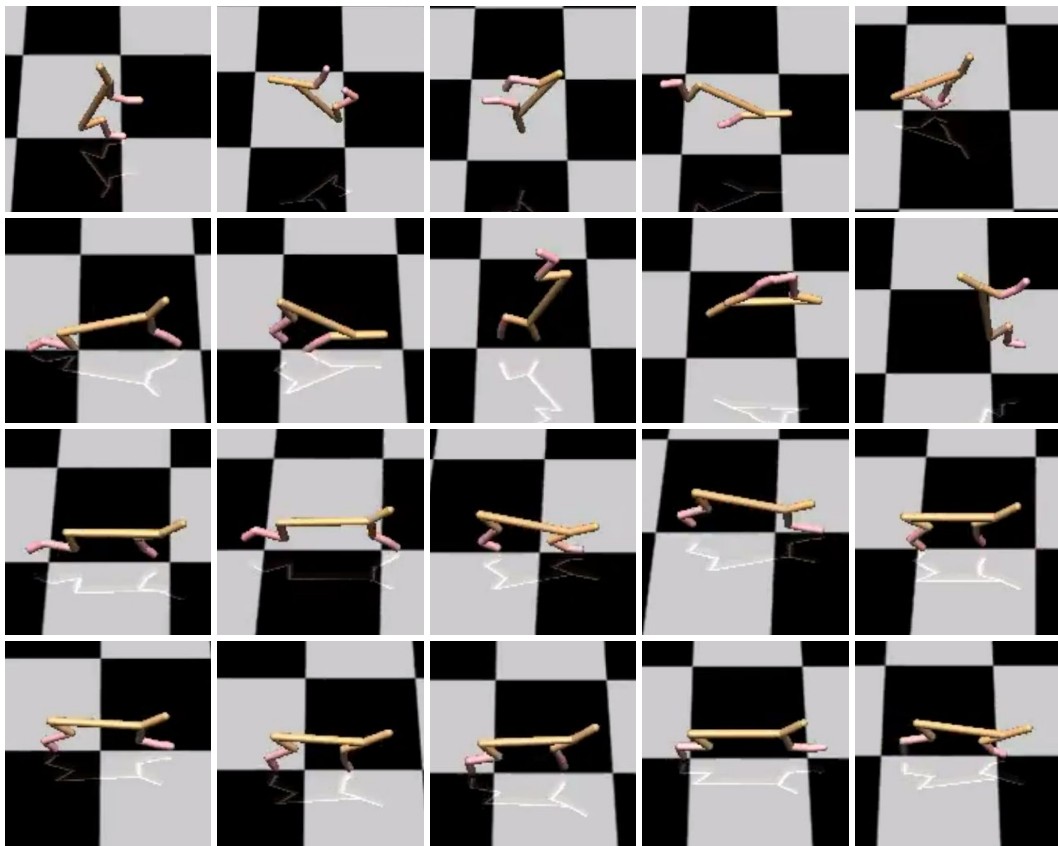

Figure 2: Examples of learned gaits. top to bottom: Back flip, Front flip, Running backwards, Running forwards.

Table 1: Improvement in performance after unsupervised learning for each method.

| Task | count | random | SU-PETS (ours) | PETS-oracle |
|---|---|---|---|---|
| cheetah-forward | 798 | 2813 | 3890 | **6238** |
| cheetah-backward | 990 | 582 | **3831** | 512 |
| cheetah-forward-flip | 1561 | 129 | **5916** | 2532 |
| cheetah-backward-flip | 699 | 369 | **5093** | 1843 |

episodes (60k time steps). Note that this number of time steps is substantially lower than the sample of complexity of deep reinforcement learning methods based on model-free learning, and was selected because it is roughly on par with the number of time steps needed for PETS alone to learn the forward running task when provided with the ground truth reward function. Recall that, during our unsupervised learning phase, no reward information whatsoever is provided to the agent. All experiments were repeated with three random seeds due to computational and time constraints, and additional seeds will be evaluated for the final.

## 4.1 COMPARISONS TO OTHER METHODS

In order to provide a comparatively evaluation of our approach, we three methods and one oracle baseline. The methods we consider are: our method, based on state uncertainty estimation, standard PETS with random exploration, and a modified version of PETS that uses the "#Exploration"

| Task | Reward Function |
|---|---|
| Running Forward | $v_{forward} - 0.1 \left\lVert a \right\rVert_2^2$ |
| Running Backwards | $-v_{forward} - 0.1 \left\lVert a \right\rVert_2^2$ |
| Back flips | $\omega - 0.1 \left\lVert a \right\rVert_2^2$ |
| Front flips | $-\omega - 0.1 \left\lVert a \right\rVert_2^2$ |

Figure 3: Reward functions.

algorithm proposed by Tang et al. (2017), which is based on estimating pseudo-counts for each

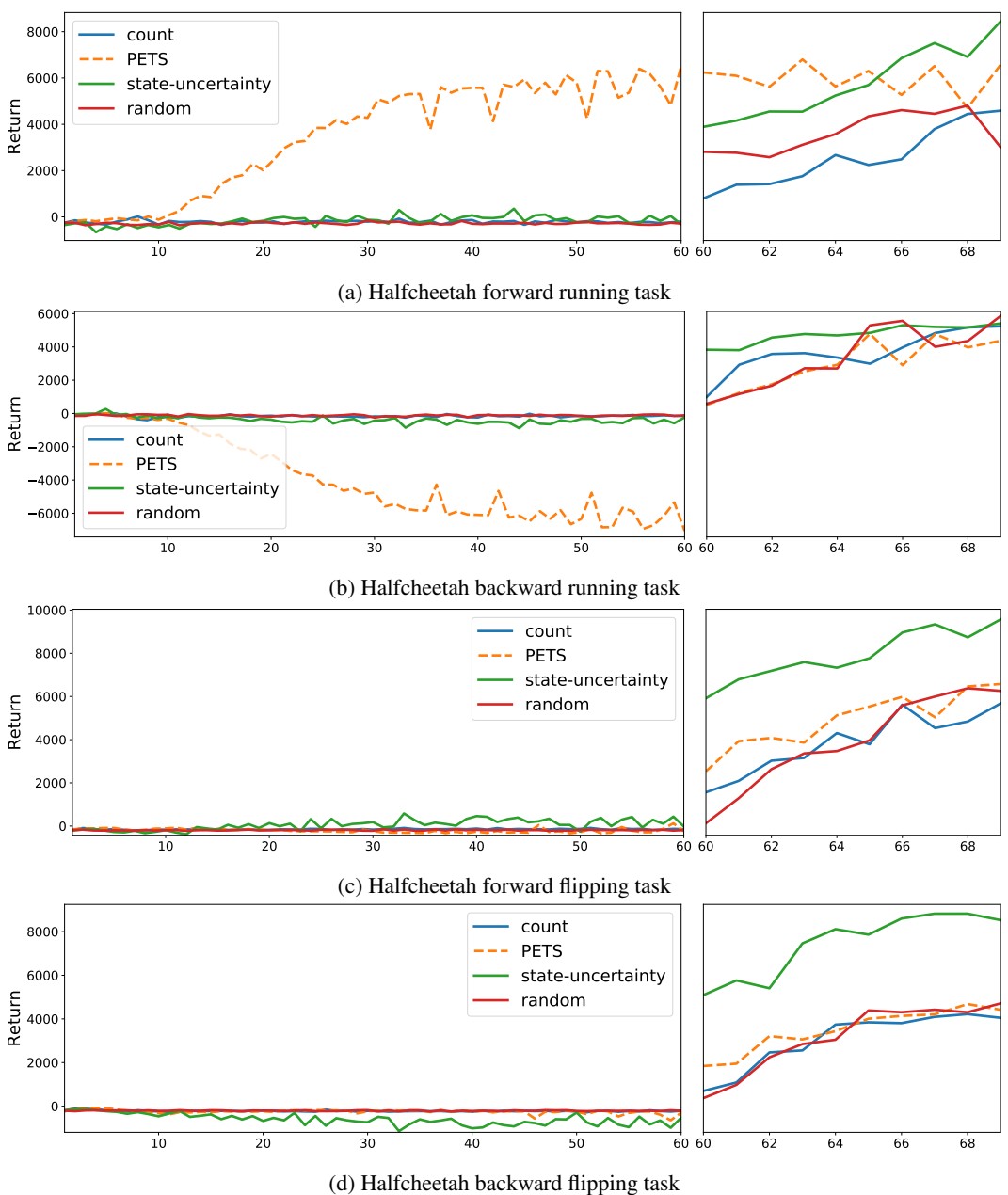

Figure 4: Results on four different Halfcheetah tasks: forwards running, backwards running, forward flipping, and backwards flipping. For each of the four experiments, we show performance during unsupervised training (left), which generally stays at zero for all methods, since they are not attempting to perform the task, except for the oracle (PETS). Once provided with the task reward (right), our method (state-uncertainty) achieves the best reward in all cases, except when compared to the oracle on the forward task.

state and visiting states with low pseudo-count. Although this algorithm is designed to provide exploration bonuses for model-free reinforcement learning, we found it to be straightforward to re-purpose to the model-based setting. We used a static hashing method as proposed by Tang et al. (2017), where the state space is discretized by using a hash function $\phi : \mathcal{S} \rightarrow \mathbb{Z}$, as done by Tang et al. (2017). Let $g : \mathcal{S} \rightarrow \mathbb{R}^D$ be any function and $A \in \mathbb{R}^{k \times D}$ a matrix whose entries are sampled from a standard Gaussian. If we let $n(z)$ denote how many states seen so far hashes to $z$, then we

define $\phi$ as

$$\phi(s) = \frac{1}{\sqrt{1 + n(\text{sgn } Ag(s))}},$$

where sgn $Ag(s) \in \{1, -1\}^k$ is interpreted as a binary integer. For our implementation, we define $g$ to be a function that normalizes $s$ according to the statistics of the current dataset. Counts are recomputed for all states with the addition of every new trajectory to account for the change in $g$. Furthermore, counts are only updated once a trajectory is completed. We then use $\phi$ in place of the previously described state uncertainty-based reward. This method is broadly representative of novelty-seeking and count-based exploration methods in the literature.

Finally, we compare to an oracle baseline, which we label as "PETS (oracle)," which consists of the PETS algorithm provided with the ground truth reward for the forward task. This oracle baseline is provided as a point of comparison for the random method, which tends to stay very close to the initial state.

The "jump-start" performance, calculated when the agent is first provided with a new task and before any additional learning has taken place, is provided in Table 1 for all methods. Naturally, the PETS (oracle) baseline performs the best on the forward task, since its "unsupervised" phase directly provides the ground truth reward for this task as input. In all other cases, our method, labeled SU-PETS, achieves substantially better performance, in many cases several times larger than the alternative approaches. We also note that the pseudo-count based method often provides a small improvement over random exploration, but not consistently. This supports the hypothesis that exploration methods designed for model-free RL do not necessarily transfer effectively to model-based RL, motivating the development of dedicated model-based exploration methods. Finally, we note that actual performance of our method on the forward HalfCheetah task, which corresponds to the standard benchmark task in the literature, is on par with good model-free methods (see, e.g., the comparisons from Haarnoja et al. (2018) and Chua et al. (2018)). This is despite the fact that the algorithm has never before seen the reward function for this task, and was provided with only 60k steps of unsupervised training – far fewer than what model-free methods typically require even when provided with the true reward function.

The plots in Figure 4 further illustrate learning curves for supervised finetuning, starting from the "jump-start" performance reported in Table 1. The horizontal axis denotes the number of trials, each of which consists of 1000 time steps. The axis starts at 60, to denote that 60 unsupervised trials were used for pre-training, but the learner did not have access to task reward during these trials. All methods improve rapidly due to the efficiency of model-based RL, but we see that the pre-trained SU-PETS can successfully incorporate even small amounts of additional experience, improving performance within just a couple of trials.

## 5 CONCLUSION

In this paper, we proposed an unsupervised reinforcement learning formulation consisting of two phases: an exploration phase where the agent can acquire a model of the environment, and an evaluation phase where it is provided with a task reward, and must either achieve high performance on this task immediately or after a short fine-tuning period. The same unsupervised exploration phase can be used across multiple downstream tasks. In order to transfer knowledge from the unsupervised phase, our method makes use of the framework of model-based RL. However, model-based RL presents two major challenges in this setting: the model-based RL procedure itself must succeed for new tasks, and the unsupervised phase must be provided with an adequate exploration objective to facilitate good state coverage, such that the learned model is accurate in wide regions of the state space. We show that both of these criteria can be fulfilled by using an uncertainty-aware model: the uncertainty estimates prevent the planner from visiting parts of the state where the model is uncertain during the second phase, and during the first phase it can be used to explicitly guide the agent to visit diverse parts of the state space.

Although transfer in reinforcement learning has been a subject of extensive study in recent years, to our knowledge our work is the first to formalize the unsupervised reinforcement learning problem in the context of model-based RL. We show that in fact this is a natural fit: the model is a task-agnostic representation of the agent's knowledge about the world, making it ideal to transfer

from the unsupervised exploration phase to downstream tasks. Our method suggests a number of promising directions for future work. First, we utilize a simple ensemble-based uncertainty estimation technique. More sophisticated methods based on Bayesian neural networks might yield better performance. Second, our fine-tuning phase naïve utilizes greedy planning – if the agents knows that it has a particular budget of trials for adaptation to a new task, these trials can also be used to explore more carefully. By improving on these dimensions of our method, future work might enable effective unsupervised reinforcement learning techniques that can solve new tasks with minimal or no training for each one.

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
