# OpenReview forum: "Unsupervised Exploration with Deep Model-Based Reinforcement Learning"
_ICLR.cc/2019/Conference_

### Official Review · AnonReviewer2 · 2018-10-31
**Decent paper, but not very novel, sparse on details.**

**Rating:** 4
**Confidence:** 4

**Review:**

The paper performs model-based reinforcement learning. It makes two main contributions. First, it divides training into two phases: the unsupervised phase for learning transition dynamics and the second phase for solving a task which comes with a particular reward signal. The scope of the paper is a good fit for ICLR.

The paper is very incremental: the ideas of using an ensemble of models to quantify uncertainty, to perform unsupervised pre-training and to explore using an intrinsic reward signal have all been known for many years.

The contribution of the paper seems to be the combination of these ideas and the way in which they are applied to RL. I have the following observations / complaints about this.

1. The paper is very sparse on details. There is no pseudocode for the main algorithm, and the quantity v^i_t (the epistemic variance on page 5) isn't defined anywhere. Without these things, it is difficult for me to say what the proposed algorithm is *exactly*.

2. Sections 1 and 2 of the paper seem unreasonably bloated, especially given the fact that the space could have been more meaningfully used as per (1).

3. The experimental section misses any kind of uncertainty estimates. If, as you say, you only had the computational resources for three runs, then you should report the results for all three. You should consider running at least one experiment for longer. This should be possible - a run of 50K steps of HalfCheetah takes about one hour on a modern 10-core PC, so this is something you should be able to do overnight.

4. The exploration mechanism is a little bit of a  mystery - it isn't concretely defined anywhere except for the fact that it uses intrinsic rewards. Again, please provide pseudocode.

As the paper states now, the lack of details makes it difficult for me to accept. However, I encourage the authors to do the following:
1. Provide pseudocode for the algorithm.
2. Provide pseudocode for exploration mechanism (unless subsumed by (1)).
3. Add uncertainty estimates to evaluation or at least report all runs.

I am willing to re-consider my decision once these things have been done.

---

> ### Comment · AnonReviewer2 · 2018-11-25
> **Update near end of discussion phase.**
>
> They authors did not address the concerns mentioned in my review, nor have they addressed the concerns of the other reviewers.
>
> In this situation, I stand by my original review.

---

> > ### Author Response · Authors · 2018-11-27
> > **Thank you**
> >
> > Dear reviewer,
> >
> > Thank you very much for your review. In response to the main criticism from all reviewers here, we have been running additional experiments on new systems and towards increasingly our method's novelty but have been unable to complete these experiments in time. We will certainty incorporate all your helpful feedback into improving a future version of this work and are grateful for the time you spent on it.

---

### Official Review · AnonReviewer3 · 2018-11-02
**An incremental work and needs more justification/clarification**

**Rating:** 4
**Confidence:** 3

**Review:**

The authors built upon the PETS algorithm to develop a state uncertainty-driven exploration strategy, for which the main point is to construct a reward function. The proposed algorithm was then tested on a specific domain to show some improvement.

The contribution of this paper may be limited, as it needs a specific setting, as shown in Figure 1. Furthermore, this paper is a bit difficult to follow, e.g., it was not until the 5th page to describe their algorithm. I summarize the pros and cons as follows.

Pros:
- The idea to include the exploration for PETS is somewhat interesting.
Cons:
- The paper is a bit difficult to follow. Just to list a few places:
  1. The term "unsupervised exploration" was mentioned a few times in this paper. I am not sure if this is an accurate term. Is there a corresponding "supervised exploration" used elsewhere?
  2. When you introduced r_t in Section 3.3, how did you use it next? Was it used in Phase II?
  3. For the PETS (oracle) in Figure 4, why are the settings different for forward and backward tasks?
  4. What does "random" mean in Figure 4?
- The novelty of this paper is somewhat limited, as it requires a specific setting and has been applied in only one domain.
- There are a few grammar mistakes/typos in this paper.
  1. What is "k" in the equation for r_t?
  2.  "...we three methods..." in Page 6.

---

> ### Author Response · Authors · 2018-11-27
> **Thank you**
>
> Dear reviewer,
>
> Thank you very much for your review. In response to the main criticism from all reviewers here, we have been running additional experiments on new systems and towards increasingly our method's novelty but have been unable to complete these experiments in time. We will certainty incorporate all your helpful feedback into improving a future version of this work and are grateful for the time you spent on it.

---

### Official Review · AnonReviewer1 · 2018-11-04
**Weak experimental evaluation and lack of novelty**

**Rating:** 4
**Confidence:** 4

**Review:**

The authors address the problem of how to use unsupervised exploration in a first phase of reinforcement learning to gather knowledge that can be transferred to new tasks to improve performance in a second task when specific reward functions are available. The authors proposed a model-based approach which uses deep neural networks as a model for the environment. The model is PETS (probabilistic ensembles with trajectory sampling), an ensemble of neural networks whose outputs parametrize predictive distributions for the next state as a function of the current state and the action applied. To collect data during the unsupervised exploration phase, they use a metric of model uncertainty computed as follows: the average over all the particles assigned to each bootstrap is computed and the variance over these computed means is the
metric of uncertainty. The authors validate their method on the HalfCheetah OpenAI gym environment where they consider 4 different tasks related to running forward, backward, tumbling forward and tumbling backward. The results obtained show that they outperform random and count based exploration approaches.

Quality:

I am concerned about the quality of the experimental evaluation of the method. The authors only consider a single environment for their experiments and artificially construct 4 relatively similar tasks. I believe this is insufficient to quantify the usefulness of the proposed method.

Clarity:

The paper is clearly written and easy to read.

Novelty:

The proposed approach seems incremental and lacks novelty. The described method for model-based exploration consists in looking at the mean of the prediction of each neural network in the ensemble and then computing the empirical average. This approach has been used before for active learning with neural networks ensembles:

Krogh, Anders, and Jesper Vedelsby. "Neural network ensembles, cross validation, and active learning." Advances in neural information processing systems. 1995.

The used model, PETS, is also not novel and the proposed methodology for having first an unsupervised learning phase and then a new specific learning task is also not very innovative.

Significance:

Given the lack of a rigorous evaluation framework and the lack of novelty of the proposed methods, I believe the significance of the contribution is very low.

---

> ### Author Response · Authors · 2018-11-27
> **Thank you**
>
> Dear reviewer,
>
> Thank you very much for your review. In response to the main criticism from all reviewers here, we have been running additional experiments on new systems and towards increasingly our method's novelty but have been unable to complete these experiments in time. We will certainty incorporate all your helpful feedback into improving a future version of this work and are grateful for the time you spent on it.

---

### Meta-Review · Area_Chair1 · 2018-12-14
**incremental, limited evaluation**

**Confidence:** 5
**Recommendation:** Reject

**Metareview:**

Strengths

The paper proposes to include exploration for the PETS (probabilistic ensembles with trajectory sampling)
approach to learning the state transition function. The paper is clearly written.

Weaknesses

All reviewers are in agreement regarding a number of key weaknesses: limited novelty, limited evaluation,
and aspects of the paper are difficult to follow or are sparse on details.
No revisions have been posted.

Summary

All reviewers are in agreement that the paper requires significant work and that it is not ready for ICLR publication.